# Aβ Chronic Exposure Promotes an Activation State of Microglia through Endocannabinoid Signalling Imbalance

**DOI:** 10.3390/ijms24076684

**Published:** 2023-04-03

**Authors:** Lucia Scipioni, Daniel Tortolani, Francesca Ciaramellano, Federico Fanti, Thais Gazzi, Manuel Sergi, Marc Nazaré, Sergio Oddi, Mauro Maccarrone

**Affiliations:** 1Department of Biotechnological and Applied Clinical Sciences, University of L’Aquila, Via Vetoio Snc, 67100 L’Aquila, Italy; 2European Center for Brain Research-IRCCS Santa Lucia Foundation (FSL), Via del Fosso di Fiorano 64, 00143 Rome, Italy; 3Department of Veterinary Medicine, University of Teramo, Via R. Balzarini 1, 64100 Teramo, Italy; 4Department of Bioscience and Technology for Food, Agriculture and Environment, University of Teramo, Via R. Balzarini 1, 64100 Teramo, Italy; 5Leibniz Research Institute for Molecular Pharmacology (FMP), Campus Berlin-Buch, 13125 Berlin, Germany; 6Department of Chemistry, Sapienza University of Rome, 00185 Rome, Italy

**Keywords:** Alzheimer’s disease, microglia, endocannabinoid system, β-amyloid peptides

## Abstract

Dysfunctional phenotype of microglia, the primary brain immune cells, may aggravate Alzheimer’s disease (AD) pathogenesis by releasing proinflammatory factors, such as nitric oxide (NO). The endocannabinoids *N*-arachidonoylethanolamine (AEA) and 2-arachidonoylglycerol (2-AG) are bioactive lipids increasingly recognised for their essential roles in regulating microglial activity both under normal and AD-driven pathological conditions. To investigate the possible impact of chronic exposure to β-amyloid peptides (Aβ) on the microglial endocannabinoid signalling, we characterised the functional expression of the endocannabinoid system on neonatal microglia isolated from wild-type and Tg2576 mice, an AD-like model, which overexpresses Aβ peptides in the developing brain. We found that Aβ-exposed microglia produced 2-fold more 2-AG than normal microglia. Accordingly, the expression levels of diacylglycerol lipase-α (DAGLα) and monoacylglycerol lipase (MAGL), the main enzymes responsible for synthesising and hydrolysing 2-AG, respectively, were consistently modified in Tg2576 microglia. Furthermore, compared to wild-type cells, transgenic microglia basally showed increased expression of the cannabinoid 2 receptor, typically upregulated in an activated proinflammatory phenotype. Indeed, following inflammatory stimulus, Aβ-exposed microglia displayed an enhanced production of NO, which was abolished by pharmacological inhibition of DAGLα. These findings suggested that exposure to Aβ polarises microglial cells towards a pro-AD phenotype, possibly by enhancing 2-AG signalling.

## 1. Introduction

Alzheimer’s disease (AD) is a neurodegenerative disease representing the primary cause of dementia in elderly people [1]. The aetiology underlying the AD neurodegenerative events remains to be fully understood, but the dysfunctional accumulation of neurotoxic β-amyloid peptides (Aβ) appears to play a pivotal role in the pathogenesis [2,3]. However, the Aβ aggregation is necessary but insufficient for developing the disease [4]. Indeed, not all aged subjects who exhibited amyloid plaques develop the disease, indicating the presence of additional processes underlying the pathogenetic evolution of amyloidosis [5].

In recent years, chronic neuroinflammation has emerged as a central cause of neuronal loss in AD, with microglial cells considered a primary player in this process [2,6,7]. Indeed, dysfunctional microglial activation, characterised by the release of proinflammatory factors and the inability to perform pro-homeostatic functions, including (i) Aβ clearance; (ii) release of neurotrophic and pro-resolution factors; and (iii) chemotaxis, induces a chronic neuroinflammatory environment that impairs the brain’s ability to repair itself [6,8].

Humoral factors and metabolic products can modulate and regulate the immune system [9,10]. Out of these, the endocannabinoids *N*-arachidonoylethanolamine (AEA) and 2-arachidonoylglycerol (2-AG) are pleiotropic bioactive lipids released “on demand” that are recognised for their essential role in regulating microglial activity under normal and AD-driven pathological conditions [10]. Indeed, these two lipids are part of a major homeostatic system, called the “endocannabinoid system” (ECS), which participates in pathophysiological processes both centrally and peripherally, with a clear impact on modulatory functions of the central nervous system [6,11]. Notably, the array of cannabinoid receptors (CB_1_ and CB_2_) and metabolic enzymes (two main biosynthetic enzymes: *N*-acylphosphatidylethanolamine-specific phospholipase D (NAPE-PLD) and diacylglycerol lipase (DAGL), for AEA and 2-AG, respectively, and two main degradative enzymes: fatty acid amide hydrolase (FAAH) and monoacylglycerol lipase (MAGL), for AEA and 2-AG, respectively [7]) are fully expressed by microglia, regulating the immune activity of these cells in terms of release of pro- and anti-inflammatory mediators, proliferation, migration, and phagocytosis [6,12]. Accumulating clinical and preclinical evidence strongly supports a profound re-arrangement of the microglial endocannabinoid signalling in AD subjects and AD-like animal models [6].

Here, we assessed the functional changes in microglial ECS induced by chronic pre-exposure to Aβ. For this aim, we isolated microglia from the brains of neonatal mice (p0-p2, namely, a presymptomatic phase, when amyloid plaques have not yet formed and there is no sign of cognitive deficits) of a murine model of amyloidosis (i.e., Tg2576) characterised by overproduction of Aβ40/42 peptides in the brain parenchyma from the earliest stages [13]. In particular, these mice express high levels of human mutant (K670N/M671L) APP transgene (hAPPK670N/M671L) under transcriptional control of the hamster prion protein promoter and develop Aβ plaques at 11–13 months of age predominantly in the neocortex and hippocampus [14]. Since microglial cells do not express the hAPP transgene [15], they cannot produce *per se* Aβ40/42 peptides once isolated and grown in tissue cultures. This allowed us to evaluate whether the pre-exposure to soluble forms of Aβ peptides, to which microglia were subjected during intrauterine life within the brain of Tg2576 mice, could result in long-term effects on the microglial phenotype and specifically on endocannabinoid signalling.

## 2. Results

### 2.1. Comparison of the Endocannabinoid Level between Wild-Type and Tg2576 Microglia

Microglia express a full array of synthetic and catabolic enzymes of the endocannabinoids and can therefore metabolise both 2-AG and AEA [12,16]. In fact, microglia produce 20 times more endocannabinoids in vitro than neurons and other glial cells and are likely the primary cellular source of these bioactive lipids under neuroinflammatory conditions [17]. Therefore, we evaluated whether chronic exposure to Aβ peptides during embryogenesis could alter the basal synthesis of endocannabinoids by microglial cells. We found that Tg2576 microglia produced ~2-fold more 2-AG than wild-type microglia (WT: 38.8 ± 8.6 pmol/10^6^ cells; Tg: 83.0 ± 28.3 pmol/10^6^ cells; *p* = 0.010). No significant difference in AEA levels was observed (Figure 1).

### 2.2. DAGLα and MAGL Were Differently Expressed in Wild-Type and Tg2576 Microglia

Since Tg2576 microglia expressed an increased level of 2-AG, we focused on evaluating the expression of the enzymes responsible for the synthesis (i.e., DAGLα and DAGLβ) and degradation (i.e., MAGL) of this endocannabinoid [7]. As shown in Figure 2, Tg2576 microglia showed a significant increase in DAGLα, measured at both mRNA and protein expression levels, which was ~4-fold higher than that of wild-type microglia (WT: 0.023 ± 0.017; Tg: 0.089 ± 0.043; *p* = 0.013). In contrast, no significant differences were found in the expression of DAGLβ, while MAGL protein expression was significantly downregulated in transgenic microglia (WT: 0.016 ± 0.006; Tg: 0.006 ± 0.003; *p* = 0.012).

### 2.3. CB_2_ Were Upregulated in Tg2576 Microglia

We then compared the expression levels of CB_1_ and CB_2_ in the two cell populations. We found no appreciable expression of CB_1_ at mRNA levels both in wild-type and Tg2576 microglia, as already described [14]. In contrast, the CB_2_ receptor was detectable in the cells, being more highly expressed at the protein level in microglial Tg2576 cells (WT: 0.360 ± 0.193; Tg: 1.461 ± 0.573; *p* = 0.004) (Figure 3a–c). To further corroborate these results, we stained wild-type and Tg2576 microglia with 0.25 μM of 8-SiR, a cell-permeable fluorescent probe for detecting endogenous CB_2_ in living cells [18]. We found that Tg2576 microglia incorporated 8-SiR, both in external and internal membranes, with faster kinetics than the wild-type cells, thus demonstrating an increased expression of the CB_2_ receptor (Figure 3, Appendix A). Image acquisition at higher magnification and resolution was performed at the end of the recording session. Upon twenty min from probe administration, cell staining remained predominantly associated with the internal membranes of microglial cells (Figure 3d).

### 2.4. Tg2576 Microglia Displayed an Enhanced Production of LPS-Induced Nitric Oxide, Which Was Reverted by Pharmacological Inhibition of DAGLα

To assess whether changes that occurred to the ECS of Tg2576 microglia (i.e., the increased expression of 2-AG and CB_2_) could alter their ability to respond to an inflammatory stimulus, we compared LPS-induced NO release between wild-type and Tg2576 microglia (Figure 4). Upon endotoxin LPS stimulation, Tg2576 microglia released 2.2-fold more NO than wild-type cells (WT_LPS: 19.2 ± 0.4 μM; Tg_LPS: 42.3 ± 9.6 μM; *p* < 0.0001), suggesting that the enhancement of CB_2_/2-AG signalling in Aβ-exposed microglia polarises these cells toward an activated phenotype. To assess whether exacerbated NO production was related to the increase in 2-AG basal level, we treated cells with KT172, a potent and selective inhibitor of DAGLα/β [19]. This inhibitor did not affect NO release in LPS-treated wild-type microglia, while it significantly reduced the NO production in LPS-treated Tg2576 microglia (Tg_LPS: 42.3 ± 9.6 μM; Tg_LPS/KT172: 26.6 ± 3.2 μM; *p* < 0.0001). Importantly, the inhibition of DAGLs completely abrogated the exacerbated release of NO by LPS-treated Tg2576 microglia that returned to the level of LPS-treated wild-type microglia (WT_LPS/KT172: 20.6 ± 1.9 μM; Tg_LPS/KT172: 26.6 ± 3.2 μM; *p* = 0.068).

## 3. Discussion

We found that primary microglia pre-exposed to Aβ displayed an increased level of CB_2_ and 2-AG, the latter being associated with an increase in DAGLα and a decrease in MAGL. Following stimulation with LPS, we also found that Tg2576 microglia showed an exacerbated NO production, which was reverted by pharmacological inhibition of DAGLα. These findings suggest that pre-exposure of detergent-soluble forms of Aβ primes microglial cells toward an “activated” phenotype by enhancing 2-AG signalling.

Several clinical and preclinical studies have revealed a strong alteration of microglial endocannabinoid signalling in AD subjects and AD-like mice [6]. In particular, CB_2_ receptors were observed to be abundantly and selectively expressed in microglia associated with neuritic plaque in the brains of AD patients [20]. Another report showed that CB_2_ mRNA expression in circulating monocytes has a positive correlation with neurological and neurocognitive severity of AD [21]. Consistently, a marked increase in CB_2_ levels was also found in microglia of AD-like mice [22,23]. Finally, ex vivo studies showed that CB_2_ mRNA was upregulated in microglia following activation with proinflammatory stimuli (interferon-γ and LPS) [24,25]. Regarding the endocannabinoid metabolism, an early report found enhanced enzymatic activity in the hippocampus of AD human patients of DAGL and MAGL [26]. In particular, increased DAGL expression within hippocampal neurons and local microglia was positively correlated with pathological AD progression in post-mortem [27]. Piro and colleagues found in the brains of AD-like mice that levels of 2-AG were considerably increased [28]. Incidentally, our findings suggest that prolonged exposure to Aβ peptides primes microglia towards an “activated phenotype”, which is the direct cause of an upregulation of CB_2_ and 2-AG signalling, as it has been observed in several clinical and preclinical studies [11].

Interestingly, there are contradictory findings on the role of 2-AG in the modulation of murine macrophage/microglia responses. On the one hand, 2-AG promotes the acquisition of an anti-inflammatory phenotype in innate immune cells [29]. On the other hand, it stimulates NO production by LPS-stimulated macrophages [30] and in microglial cells (our findings). Incidentally, it has been suggested that these discrepancies in the effects of 2-AG could be due to its conversion into arachidonic acid, and then to its bioactive COX-2 metabolites [31].

Our study showed that microglial CB_2_ is predominantly localized in intracellular compartments, reminiscent of the endoplasmic reticulum and Golgi apparatus. These findings are in full agreement with those already described in other cellular models, such as neurons and leukocytes [14,18,32]. This aspect seems to be a feature that distinguishes CB_2_ from CB_1_, whose functional expression is mostly confined to the plasma membrane of cells [14]. However, the question of whether the CB_2_ receptor present within the cell is functional requires further investigation.

Finally, we have not studied the molecular mechanisms by which amyloidosis influences the expression of ECS elements. However, it is not surprising that the pre-exposure of microglial cells to soluble forms of Aβ peptides during their intrauterine life stimulates the microglial endocannabinoid signalling. As a matter of fact, the ECS is a pro-homeostatic system activated by various external and/or internal stimuli, eliciting multiple adaptive responses, including neuromodulation and immunomodulation that help to maintain and/or restore brain homeostasis [6]. In this context, Aβ peptides represent the internal insults that trigger the activation of this lipid signalling, possibly through toll-like receptor-dependent mechanisms.

In conclusion, this investigation shows that prolonged exposure to Aβ peptides leads to the acquisition of a pre-activated state by microglial cells, making them more prone to responding to inflammatory stimuli via 2-AG-dependent signalling. Although further studies are needed to elucidate the molecular mechanisms by which Aβ peptides influence the ECS, these Aβ-induced changes in the endocannabinoid signalling may contribute to acquiring a dysfunctional phenotype of microglial cells throughout the progression of AD-driven pathological conditions.

## 4. Materials and Methods

### 4.1. Reagents

Chemicals were of the purest analytical grade. DMEM, foetal bovine serum (FBS), and other cell culture reagents were purchased from Corning (Corning, New York, NY, USA). AEA, 2-AG, *N*-arachidonoylethanolamine-d8 (AEA-d8), and 2-arachidonoylglicerol-d8 (2-AG-d8) were purchased from Cayman Chemicals (Ann Arbor, MI, USA). KT172 and LPS from *Escherichia coli* O111:B4 were purchased Sigma-Aldrich (St. Louis, MO, USA). All other chemicals were purchased from Sigma-Aldrich, unless stated otherwise.

### 4.2. Mice

Tg2576 mice were purchased from Taconic Biosciences (strain: B6;SJL-Tg(APPSWE)2576Kha; cod. 1349-M). A colony of Tg2576 was obtained by crossing heterozygous transgenic male mice with C57BL/6J x SJL hybrid female mice, as previously described [13]. For colony genotyping, tail tissues were sampled from in order to extract genomic DNA, according to standard procedures. DNA yield and purity were assessed by evaluating 260/280 nm and 260/230 nm absorbance ratios using a Nanodrop spectrophotometer (Thermo Fisher Scientific, Milan, Italy). An amount of 10 ng of DNA per sample was amplified by means of polymerase chain reaction (PCR): all assays were carried out using specific primers (FW 5′-CTG ACC ACT CGA CCA GGT TCT GGG T-3′, REV 5′-GTG GAT AAC CCC TCC CCC AGC CTA GAC CA-3′, Sigma-Aldrich (St. Louis, MO, USA)). The thermal profile was a 1′ initial denaturation step at 95 °C followed by 35 cycles of amplification (denaturation: 15″ at 95 °C; annealing: 15″ at 55 °C; and extension: 10″ at 72 °C). Genotyping was later performed by running amplified DNA samples on a 2% agarose gel (Sigma-Aldrich) in presence of 0.01% of GelRedTM nucleic acid staining (Biotium, Hayward, CA, USA). Tg2576 mice were identified by the presence of a ≈500 bp band.

### 4.3. Neonatal Microglia Cell Cultures and Treatment

Neonatal microglia cells were obtained from mixed glia cultures derived from the cerebral cortices of post-natal day 0–2 (p0–p2) C57BL/SJL wild-type and Tg2576 mice, as described [33]. After 5–7 days, cells were collected after vigorously taping the flask. These procedures gave almost pure (no more than 2% contamination) microglial cell population.

### 4.4. Measurements of Endogenous Levels of the Endocannabinoids by Liquid Chromatography-Mass Spectrometry

AEA and 2-AG were evaluated in cell samples by ultra-high-performance liquid chromatography-tandem mass spectrometry [34]. Briefly, the lipid fraction from primary microglia was extracted using chloroform–methanol–water (2:1:1 *v*/*v*) in the presence of internal standards (i.e., 1 ng/mL of AEA-d8, 100 ng/mL of 2-AG-d8). The organic phase was dried under a gentle nitrogen stream and then subjected to a micro-solid phase extraction (μSPE) procedure for a rapid clean-up using OMIX C18 tips from Agilent Technologies (Santa Clara, CA, USA). All analyses were performed using a Nexera XR LC 20 AD UHPLC system (Shimadzu Scientific Instruments, Columbia, MD, USA) that was equipped with Kinetex XB-C18 1.7 μm 100 × 2.1 mm from Phenomenex (Torrance, CA, USA) and coupled with a 4500 Qtrap from Sciex (Toronto, ON, Canada) that was equipped with a Turbo V electrospray ionization source. The levels of AEA and 2-AG were then calculated as pmoles/cell number.

### 4.5. qRT-PCR

Total RNA was extracted with a ReliaPrep RNA Miniprep System kit (Promega, Milan, Italy). SuperScript IV VILO Reverse Transcriptase (Invitrogen, Waltham, MA, USA) was used for cDNA synthesis. Transcripts were quantified by real-time quantitative PCR on a StepOne Real-Time PCR System sequence detector (Applied Biosystems, Life Technologies, Carlsbad, CA, USA) with Applied Biosystems predesigned TaqMan Gene Expression Assays. The following probes were used (Applied Biosystems, Life Technologies, Carlsbad, CA, USA, assay identification numbers in parentheses): Actin-ꞵ (Mm04394036_g1), DAGLɑ (Mm00813830_m1), DAGLꞵ (Mm00523381_m1), and MGLL (Mm00449274_m1).

### 4.6. Western Blotting

Western blotting was performed according to standard procedures. Briefly, cellular pellets were resuspended in a suitable volume of ice-cold lysis buffer containing 50 mM Tish pH 7.4, 150 mM NaCl, 1 mM EDTA, 1% TritonX-100, and a mixture of protease and phosphatase inhibitors, and then were sonicated in the ice-cold lysis buffer for 1 min continuous pulses. The protein concentration was measured using a Bradford assay kit (Bio-Rad, Hercules, CA, USA). Equal aliquots (30 μg/lane) of each sample were denatured for 10 min at 95 °C separated by SDS-PAGE (10%) and transferred to nitrocellulose membranes. Blots were then incubated with primary antibodies recognizing mouse anti-actin (1:10,000, Sigma-Aldrich, St. Louis, MO, USA, cat. no. A5441), mouse anti-DAGLɑ (1:500, Santa Cruz Biotechnology Inc., Dallas, TX, USA, cat. no. sc-390409), rabbit anti-DAGLꞵ (1:500, Thermo Fisher, Waltham, MA, USA, cat. no. PA5-26331), and rabbit anti-MAGL (1:500, Abcam, Cambridge, UK, cat. no. ab24701). After incubation with the appropriate peroxidase-conjugated antibody (Santa Cruz Biotechnology Inc., Dallas, TX, USA), blots were developed using an enhanced chemiluminescence detection system Lightwave Plus, (GVS, Bologna, Italy). Chemiluminescence signals were detected by exposing blots to the C-DiGit blot scanner (LI-COR, Lincoln, NE, USA). C-DiGit images were analyzed by using Image Studio Software 4.0.21 (LI-COR). Densities of protein bands in the Western blots were measured, and mean ratios between proteins and actin were reported.

### 4.7. Immunofluorescence and Confocal Analysis

Cell surface CB_2_ receptors were assessed by live confocal imaging using 8-SiR, a highly potent, fluorescent CB_2_ agonist probe, as previously described [18]. Briefly, primary wild-type and Tg2576 microglia were seeded at the density of 2 × 10^4^ cell/mL on the wells of chambered coverslips (Ibidi, Gräfelfing, Germany) and cultured for 24 h. Cells were incubated with 0.2 μM 8-SiR and time-lapse imaging was performed by using a confocal laser scanning microscope ZEISS LSM 800 equipped with an Airyscan detection unit (Zeiss, Oberkochen, Germany). Live imaging was performed at 22 ± 2 °C by recording one frame every 15 sec for 20 min. At the end of the recording session, living cells were imaged with Airyscan mode. Each image was taken at the equatorial plane of the cells using the ZEN Blue 2.3 software (Zeiss, Oberkochen, Germany). Super-resolution image processing was performed using the Airyscan processing toolbox in the ZEN software. The data were exported as TIFF files and analyzed using the Fiji software (National Institutes of Health; version 2.3.0/1.53f (13 September 2021); https://imagej.net/Fiji). A Gaussian kernel filter was applied to the images using a standard deviation of 0.8 pixels. All intensity profiles were background subtracted and normalized to the frame taken at the end of the registration. For presentation purposes, images were exported in Artstudio Pro version 2.0.13 (Lucky Clan, Lodz, Poland; http://www.luckyclan.com) for adjustments of brightness and contrast.

### 4.8. Measurement of Nitric Oxide

For the measurement of NO, cells were seeded (3 × 10^4^/well) into 96-well plates in 100 µL of DMEM and were left untreated or treated for 20 h with 100 ng/mL LPS and 0.1 µM KT172, used alone or in combination. NO levels were determined in thawed supernatants by measuring the amount of nitrite, a stable metabolic product of NO, according to the reported procedure [35]. Briefly, the absorbance of the assay mixture, which contained medium (100 μL) plus Griess reagent (100 μL) in round-bottom 96-well tissue culture plates, was measured at 570 nm using a microplate reader (Thermo Fisher, Waltham, MA, USA). Sodium nitrate solutions (0.05–10 μM) were used for preparing calibration curves.

### 4.9. Statistical Analysis

All results are reported as mean ± standard deviation (S.D.). Data were elaborated and analysed statistically using the R Statistical Package (R version 4.2.2 (31 October 2022)) within RStudio software (2022.12.0 + 353 version: https://rstudio.com/) or the GraphPad Prism (version 9). In accordance with the European recommendation 2010/63/EU regarding the protection of laboratory animals and the ARRIVE guidelines established by the NC3Rs (National Center for the Replacement, Refinement and Reduction of Animals in Research), an *a priori* power analysis was conducted using G*Power version 3.1.9.6 for sample size estimation based on data from a pilot study. The data were first tested for normality (Shapiro–Wilk test), and then analysed by an unpaired Welch’s *t*-test with Holm-Sidak correction for multiple comparisons. NO levels were analysed using a two-way ANOVA with genotype and treatment as the main factors. Post hoc tests were made using Tukey’s test for multiple comparisons. Differences were considered significant at the *p* < 0.05 level. The statistical methods used for each analysis are specified in the figure legends.

## Figures and Tables

**Figure 1 ijms-24-06684-f001:**
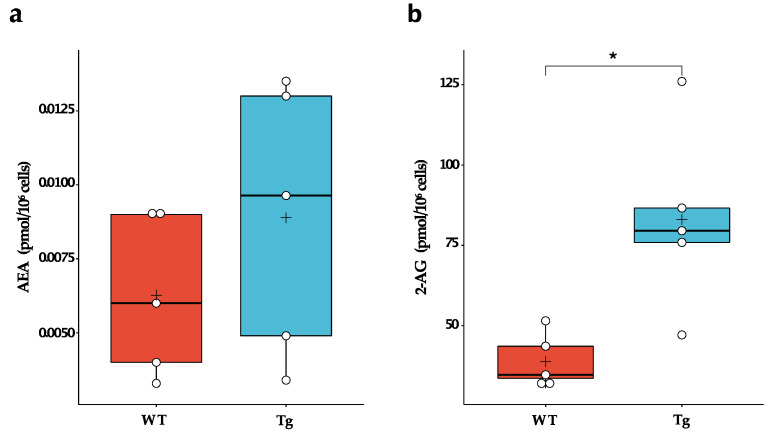
Endogenous levels of endocannabinoids in wild-type (WT) and Tg2576 (Tg) microglia. (**a**) AEA and (**b**) 2-AG basal levels in WT and Tg microglia were quantified by liquid chromatography-mass spectrometry. Horizontal line and + within the rectangle (boxplot) represent the median and the mean, respectively, of the values of five independent experiments (white circles). Significance is shown as *p* value, calculated using an unpaired *t*-test. * *p* < 0.05.

**Figure 2 ijms-24-06684-f002:**
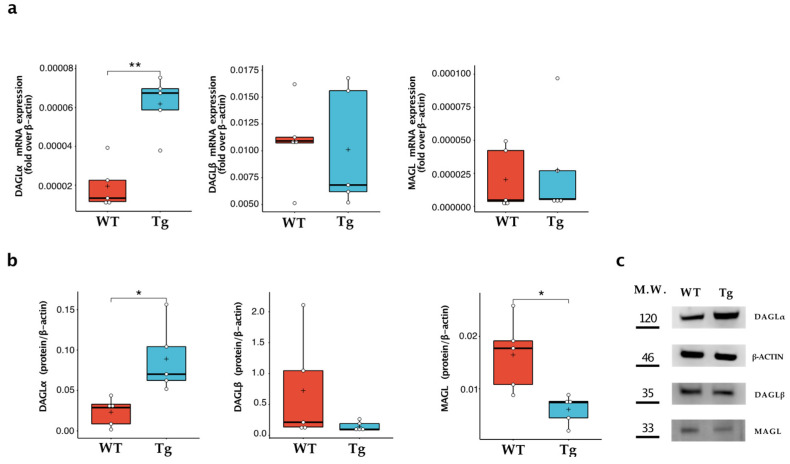
Characterisation of 2-AG metabolic enzymes in wild-type microglia (WT) and Tg2576 microglia (Tg). Enzymes of synthesis (i.e., DAGLα and DAGLβ) and degradation (i.e., MAGL) were quantified (**a**) for their mRNA content by quantitative real-time polymerase chain reaction and (**b**) for their protein expression by Western blot analysis. Horizontal line and + within the rectangle (boxplot) represent the median and the mean, respectively, of the values of five independent experiments (white circles). Significance is shown as a *p* value, calculated using an unpaired *t*-test. * *p* < 0.05; ** *p* < 0.01. (**c**) Representative immunoblot showing the expression of the indicated protein.

**Figure 3 ijms-24-06684-f003:**
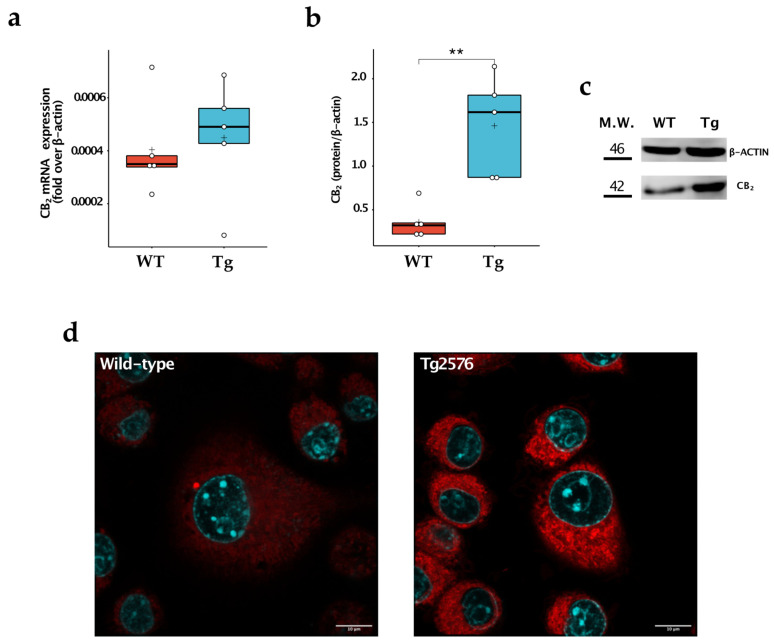
Study of CB_2_ expression in wild-type (WT) and Tg2576 (Tg) microglia. (**a**) CB_2_ was quantified for its mRNA content by quantitative real-time polymerase chain reaction and (**b**) for its protein expression by Western blot analysis. Horizontal line and + within the rectangle (boxplot) represent the median and the mean, respectively, of the values of five independent experiments (white circles). Significance is shown as a *p* value, calculated using an unpaired t-test. ** *p* < 0.01. (**c**) Representative immunoblot showing the expression of the indicated protein. (**d**) Super-resolution fluorescence microscopy in microglial cells using probe 8-SiR. Cells were pre-stained with Hoechst 33,342 (cyan) to counterstain the nuclei and then incubated for 20 min with 0.25 μM 8-SiR (red). See also AVI Appendix A.

**Figure 4 ijms-24-06684-f004:**
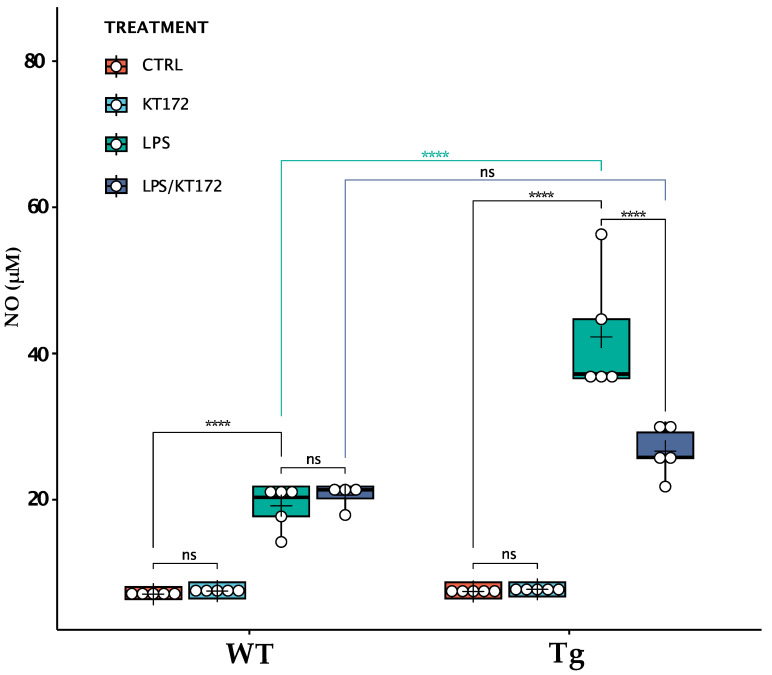
Release of NO by wild-type (WT) and Tg2576 (Tg) microglia following different treatments. WT and Tg2576 microglia were left untreated (CTRL) or were treated with 0.1 µM KT172 (KT172), 100 ng/mL LPS (LPS), or with their combination (LPS/KT172) for 20 h. After each treatment, NO content was measured in cell supernatants using a Griess assay. Horizontal line and + within the rectangle (boxplot) represent the median and the mean, respectively, of the values of five independent experiments (white circles). A two-way ANOVA was used to compare differences between different treatments in microglia of different genotypes. Significance is shown as a *p* value, calculated using a post hoc Tukey’s test. **** *p* < 0.0001; ns: not significant.

## Data Availability

Data available in a publicly accessible repository that does not issue DOIs. This data can be found here: [https://docs.google.com/spreadsheets/d/1WDWqlu1bTOljJ9stgt_zyQclW8hvW6kwSCeORXhsQck/edit?usp=share_link].

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
