# Peer review of "Aβ Chronic Exposure Promotes an Activation State of Microglia through Endocannabinoid Signalling Imbalance"

_ijms, 2023, doi:10.3390/ijms24076684_

Round 1
Reviewer 1 Report
In this manuscript Scipioni et al. have studied the consequence of chronic Aβ exposure on the activation state of microglia. The authors demonstrate an increase in the levels of 2-arachidonoylglycerol (2-AG) in microglia isolated from Tg2576 mice, a mouse model characterized by excess Aβ without cognitive decline in early stages. They attribute this increase in 2-AG to the high mRNA and protein levels of DAGL, the principal enzyme responsible for producing 2-AG. The authors also demonstrate increased protein and mRNA levels of CB2 in Tg2576 microglia compared to WT controls. Finally, the authors show that LPS-driven nitrous oxide production is higher in Tg2576 microglia (compared to WT) and that this can be rescued by treating cells with KT172, an inhibitor of the enzyme DAGL. Therefore, the authors conclude that upon exposure with Aβ, microglia go into activated state due to endocannabinoid signaling imbalance.
The manuscript is well structured, and the authors systematically answer important scientific questions. The authors provide novel data of microglial activation due to endocannabinoid signaling. However, some concerns need to be addressed (see below).
1. Since the biggest claim of the authors is that the endocannabinoid signaling imbalance observed in Tg2576 microglia is due to excessive Aβ peptide production, it is important that they demonstrate increased production of Aβ, specifically in the microglia, compared to WT controls (especially considering the authors do not provide a citation for this statement- “At this aim, we isolated microglia from the brains of neonatal 67 mice (p0-p2, namely, a presymptomatic phase, when amyloid plaques have not yet 68 formed and there is no sign of cognitive deficits) of a murine model of amyloidosis (i.e., 69 Tg2576) characterised by overproduction of Aβ40/42 peptides in the brain parenchyma 70 from the earliest stages”
2. In figure 4, the authors demonstrate, very clearly, that KT172 treatment suppresses LPS-driven nitrous oxide production in Tg2576 microglia. It would be interesting to see what happens to DAGL levels, since there was a clear upregulation of DAGL in these microglia. Furthermore, do 2-AG levels also reduce to WT levels?
3. On similar lines to point 2, what about CB2 levels upon KT172 treatment? Since the authors attribute microglial activation due to cannabinoid dysfunction, it would be important to demonstrate that CB2 levels are also restored upon KT172 treatment,
Author Response
First of all, we would like to thank both Reviewers for the important and constructive comments, which we found very helpful indeed, in order to improve the clarity and impact of our work. Please note that all changes have been highlighted in red in the revised version. Point-by-point replies to the criticism raised are detailed below.
Reviewer #1
In this manuscript Scipioni et al. have studied the consequence of chronic Aβ exposure on the activation state of microglia. The authors demonstrate an increase in the levels of 2-arachidonoylglycerol (2-AG) in microglia isolated from Tg2576 mice, a mouse model characterized by excess Aβ without cognitive decline in early stages. They attribute this increase in 2-AG to the high mRNA and protein levels of DAGL, the principal enzyme responsible for producing 2-AG. The authors also demonstrate increased protein and mRNA levels of CB2 in Tg2576 microglia compared to WT controls. Finally, the authors show that LPS-driven nitrous oxide production is higher in Tg2576 microglia (compared to WT) and that this can be rescued by treating cells with KT172, an inhibitor of the enzyme DAGL. Therefore, the authors conclude that upon exposure with Aβ, microglia go into activated state due to endocannabinoid signaling imbalance.
The manuscript is well structured, and the authors systematically answer important scientific questions. The authors provide novel data of microglial activation due to endocannabinoid signaling. However, some concerns need to be addressed (see below).
- Since the biggest claim of the authors is that the endocannabinoid signaling imbalance observed in Tg2576 microglia is due to excessive Aβ peptide production, it is important that they demonstrate increased production of Aβ, specifically in the microglia, compared to WT controls (especially considering the authors do not provide a citation for this statement- “At this aim, we isolated microglia from the brains of neonatal 67 mice (p0-p2, namely, a presymptomatic phase, when amyloid plaques have not yet 68 formed and there is no sign of cognitive deficits) of a murine model of amyloidosis (i.e., 69 Tg2576) characterised by overproduction of Aβ40/42 peptides in the brain parenchyma 70 from the earliest stages”
R: We thank the Reviewer for the valuable comments. We want to emphasize that, in our AD-like model (Tg2576 mice), the production of Aβ40/42 peptides is restricted to neurons (please, see ref. 15). Indeed, Heiland and colleagues demonstrated that the primary microglia obtained from Tg2576 mice do not express the hAPP, so they cannot produce Aβ40/42 peptides. We aimed to assess the impact of pre-exposure to Aβ peptides, that microglia experience during embryogenesis in Tg2576 mice, on the functional changes in the eCB signalling of these cells. In this context, it is essential to note that the detergent-soluble form of Aβ is present in the brain parenchyma throughout the life of these mice, whereas the detergent-insoluble form of Aβ is absent until six months of age (Westerman A., et al; The Relationship between Aβ and Memory in the Tg2576 Mouse Model of Alzheimer’s Disease; The Journal of Neuroscience, March 1, 2002, 22(5):1858–1867; Maccarrone et al., Pharmacol Res 2018, 130, 366–373, doi:10.1016/j.phrs.2018.02.009.). These points are now better described and discussed in the revised text (see page 2; lines 81-86).
- In figure 4, the authors demonstrate, very clearly, that KT172 treatment suppresses LPS-driven nitrous oxide production in Tg2576 microglia. It would be interesting to see what happens to DAGL levels, since there was a clear upregulation of DAGL in these microglia. Furthermore, do 2-AG levels also reduce to WT levels?
R: We appreciated the suggestion of the Reviewer. As a matter of fact, KT172 is a potent inhibitor of the enzymatic activity of DAGL and that we do not expect an indirect effect on DAGL gene expression (Buczynski et al., PNAS 26;113(4):1086-91). Regarding the second issue, as a potent DAGL inhibitor it is plausible that KT172 robustly reduces basal 2-AG levels in both wild-type and Tg2576 cells. Unfortunately, we cannot confirm this assumption with further experiments because the procedures needed (namely, murine pairings and isolation and culturing of primary microglia) are quite complex and time-consuming (at least two months), so it is not possible to perform them by the resubmission deadline.
- On similar lines to point 2, what about CB2 levels upon KT172 treatment? Since the authors attribute microglial activation due to cannabinoid dysfunction, it would be important to demonstrate that CB2 levels are also restored upon KT172 treatment,
R: As reported above, we do not expect that inhibition of DAGLs by KT172 revert the CB2 overexpression, which is indeed attributable to previous exposure of microglial cells to amyloid peptides. However, we cannot exclude the existence of a feed-back regulation between 2-AG and its receptor. Again, for the reasons stated above, we cannot test this possibility and believe that this aspect deserves to be investigated with a more dedicated study.
Reviewer 2 Report
The authors aimed to investigate the possible impact of chronic exposure to Aβ on the microglial endocannabinoid signalling. They characterised the functional expression of the endocannabinoid system on neonatal microglia isolated from wild-type and Tg2576 mice, an AD-like model. Aβ-exposed microglia were found produced two-fold more 2-AG than normal microglia. The findings suggested that exposure to Aβ polarises microglial cells towards a pro-AD phenotype, possibly by enhancing 2-AG signalling. The research content is generally innovative, but there are also some problems.
Figure 1: The dispersion of the results is relatively large, the results are difficult to be reliable, and the large sample size is missing.
Figure 2: Representative immunoblot showing the expression of the indicated protein. The result seems not as good as A and C. Please provide all the original immunoblot.
The results of the article are poor, and the current form is not enough for publication.
Author Response
Replies to the Reviewer’comments
First of all, we would like to thank both Reviewers for the important and constructive comments, which we found very helpful indeed, in order to improve the clarity and impact of our work. Please note that all changes have been highlighted in red in the revised version. Point-by-point replies to the criticism raised are detailed below.
Reviewer #2
The authors aimed to investigate the possible impact of chronic exposure to Aβ on the microglial endocannabinoid signalling. They characterised the functional expression of the endocannabinoid system on neonatal microglia isolated from wild-type and Tg2576 mice, an AD-like model. Aβ-exposed microglia were found produced two-fold more 2-AG than normal microglia. The findings suggested that exposure to Aβ polarises microglial cells towards a pro-AD phenotype, possibly by enhancing 2-AG signalling. The research content is generally innovative, but there are also some problems.
- Figure 1: The dispersion of the results is relatively large, the results are difficult to be reliable, and the large sample size is missing.
R: We thank the Reviewer for the useful comments. The number of animals used in our study is based on statistical analyses (a priori power analysis to calculate sample sizes) in accordance with the European recommendation 2010/63/EU regarding the protection of laboratory animals and the ARRIVE guidelines established by the NC3Rs (National Center for the Replacement, Refinement and Reduction of Animals in Research). We clearly stated this fact in the revised manuscript.
Although it may seem that the sample size is not sufficiently large, given the relative variability observed experimentally, it is important to note that the study still found significant differences, as well as covered all levels of biology of the different elements considered: from the levels of the lipids (Fig.1) to those of mRNAs (Fig. 2a) and proteins (Fig. 2b) of their metabolic enzymes. We therefore believe that the overall results are solid as well as reciprocally consistent.
Figure 2: Representative immunoblot showing the expression of the indicated protein. The result seems not as good as A and C. Please provide all the original immunoblot.
R: Please note that panel represents the mRNA expression measured by RT-PCR, while panel b represents the protein expression measured by WBs, one of which is shown in panel c. In this regard, we would like to point out that we have indeed already provided all the immunoblots, as required during the submission process.
The results of the article are poor, and the current form is not enough for publication.
Round 2
Reviewer 2 Report
accepted.